# Educommunication in Nutrition and Neurodegenerative Diseases: A Scoping Review

**DOI:** 10.3390/ijerph21081113

**Published:** 2024-08-22

**Authors:** Karla Mônica Dantas Coutinho, Sancha Helena de Lima Vale, Manacés dos Santos Bezerril, Mônica Karina Santos Reis, Almudena Muñoz Gallego, Karilany Dantas Coutinho, Ricardo Valentim, Lucia Leite-Lais, Kenio Costa de Lima

**Affiliations:** 1Postgraduate Program in Health Sciences, Federal University of Rio Grande do Norte, Natal 59012-300, Brazil; 2Laboratory of Technological Innovation in Health (LAIS), Federal University of Rio Grande do Norte, Natal 59012-300, Brazilkarilany@lais.huol.ufrn.br (K.D.C.);; 3Department of Nutrition, Federal University of Rio Grande do Norte, Natal 59078-970, Brazil; 4Postgraduate Program in Nursing, Federal University of Rio Grande do Norte, Natal 59012-300, Brazil; 5Department of Odontology, Federal University of Rio Grande do Norte, Natal 59064-630, Brazil; 6Department of Theories and Analysis of Communication, Complutense University of Madrid, 28040 Madrid, Spain; 7Department of Biomedical Engineering, Federal University of Rio Grande do Norte, Natal 59066-800, Brazil; 8Postgraduate Program in Management and Innovation and Health, Federal University of Rio Grande do Norte, Natal 59066-800, Brazil

**Keywords:** neurodegenerative diseases, education, communication, nutrition

## Abstract

Neurodegenerative diseases significantly impact individuals’ nutritional status. Therefore, nutritional education plays a crucial role in enhancing the understanding of food and nutrition, preventing or minimizing malnutrition, promoting well-being, and empowering patients and caregivers. Educommunication is a methodology that utilizes communication as a pedagogical tool, with the potential to positively enhance the teaching–learning process. This study aims to identify and map educommunication strategies designed to educate caregivers and patients with neurodegenerative diseases about food and nutrition. Methods: This scoping review followed the JBI Institute Reviewer’s Manual. The search was conducted between June 2022 and March 2023 in databases including PubMed/MEDLINE, Embase, Scopus, and Web of Science. Results: Out of 189 studies identified, 29 met the eligibility criteria, and only 3 were suitable for inclusion in this review. Conclusion: Studies using educommunication for food and nutrition education are scarce. Despite the limited number of studies included in this review, various educommunication strategies utilizing communication and information technologies were used. Educommunication strategies can facilitate knowledge acquisition in food and nutrition and change behaviors, resulting in health benefits for the participants. More studies on this subject are needed.

## 1. Introduction

Neurodegenerative diseases are characterized by the degeneration and/or death of neurons, leading to the progressive loss of motor, physiological, neuropsychological, and cognitive abilities, directly affecting the quality of life [1,2]. Neurodegenerative diseases encompass various diseases [3], with the most prevalent being Alzheimer’s Disease (AD), Parkinson’s Disease (PD), Multiple Sclerosis (MS), and Motor Neuron Diseases (MND) [4]. Although multifactorial interactions are evident, nutrition plays a significant role in the pathogenesis and progression of these diseases [5].

Neurological diseases are often associated with malnutrition and dysphagia, which have a major impact on the individual’s nutritional status. Oropharyngeal dysphagia is one of the main causes of malnutrition in patients with neurodegenerative diseases, as it impairs food intake and oral hydration. Weight loss and malnutrition can worsen neurodegenerative processes, accelerate the progression of neurodegenerative diseases, and lead to a poorer prognosis. This often reduces quality of life and increases mortality in affected patients [5,6]. Thus, acquiring knowledge about safe food intake is fundamental to building autonomy in care and improving the quality of life and survival of these patients [7], since adequate nutrition can mitigate the progression of neurodegenerative diseases [8].

Food and nutrition education for caregivers and patients with neurodegenerative diseases aims to promote autonomous and voluntary practice. Through various educational strategies it is possible to improve knowledge and behavior related to nutrition, reflecting benefits to health and well-being [9]. One of the strategies used in food and nutrition education is educommunication

Educommunication is a pedagogical methodology that originated in Latin America during the 1960s and 1970s. It represents a blend of education and communication designed to foster an open, critical, and creative environment for learning ecosystems [10,11,12,13]. Educommunication is distinct from other forms of communication in medical/health education due to its unique approach and objectives. According to Soares’ perspective [14], the core characteristics of educommunication, includes:(a)Integration of Education and Communication: Educommunication combines educational and communicative practices to create a holistic learning environment. This methodology seeks to facilitate social dialogue using information and communication technologies, favoring the development of enriching pedagogical action strategies [14,15]. Unlike traditional methods that may rely solely on one-way information delivery (e.g., lectures), educommunication emphasizes interactive, dialogical processes.(b)Open, Critical, and Creative Ecosystems: The methodology encourages the construction of educational content, uses simple language, and seeks to intervene to expand the awareness and critical participation of individuals in communicative ecosystems [16]. This approach contrasts with more rigid educational frameworks by promoting critical thinking and creativity among participants.(c)Use of Information and Communication Technologies (ICTs): Educommunication leverages various digital tools such as open educational resources, videos, and podcasts to facilitate learning [17]. This integration of technology helps make educational content more accessible and relevant to individuals’ daily lives.(d)Facilitation of Social Dialogue: Educommunication aims to enhance social dialogue, promoting discussions and exchanges between participants. This is different from traditional medical education, which might focus more on one-way dissemination of information rather than interactive dialogue [14,18,19].(e)Empowerment and Autonomy: Educommunication seeks to empower learners by enhancing their ability to express themselves and participate actively in their learning process. This contrasts with more conventional methods that may position learners as passive recipients of information [19].

To achieve the proposed objectives, the educommunicator can use a variety of resources, either individually or in combination, allowing tailored decision-making that meets the specific needs of the population and achieves desired results [20]. Thus, the role of the educommunicator is to plan, create, implement, and evaluate actions within the specified areas of intervention. This helps to ensure that communicative ecosystems maintain a high level of dialog among the subjects involved [14,18,19].

In educommunication, interventions raise participants’ awareness about the role of communication and dialogue in knowledge creation for better health or living conditions. These interventions empower participants, enhance their ability to express themselves, and stimulate learning. They also encourage the integration of information technologies into daily life [20].

Given the above, this scoping review aims to identify and map educommunication actions as a food and nutrition education strategy applied to caregivers and patients with neurological diseases. It also aims to identify facilitating aspects and barriers in the development and implementation of the educommunication actions proposed and published.

## 2. Methods

This scoping review was carried out according to a protocol, registered on the Home OSF platform in July 2022 (https://osf.io/zgq4x/, accessed on 21 August 2024) and published recently [21]. The preparation of this review was guided by the recommendations of the JBI Institute Reviewer’s Manual [22,23], originally proposed by Arksey and O’Malley (2005) [24]. It also followed the Preferred Reporting Items for Systematic Reviews and Meta-Analyses extension for Scoping Reviews (PRISMA ScR) [25]. The PRISMA-ScR Checklist is available as Appendix A.

### 2.1. Inclusion and Exclusion Criteria

We included observational studies, experimental studies, dissertations, and theses, available in full, which addressed educommunication as a food and nutrition education strategy applied to caregivers and patients with neurodegenerative diseases in any environment, including educational institutions, community centers, hospitals, health units, and home environments, in face-to-face or virtual formats. No time and language re-strictions were applied to our survey. Duplicate articles, editorials, manuals, books, experience reports, reflection studies, studies without abstracts, article protocols, and literature reviews were excluded.

### 2.2. Search Strategy

A search strategy was carried out between June 2022 and March 2023, in the following databases: PubMed/MEDLINE, Excerpta Medica Database (Embase), Scopus, and Web of Science. In addition, the electronic library ‘SciELO’ and the web search engine ‘Google Scholar’ were also used in our search. All records retrieved were peer-reviewed. The search terms and equations for each database are described in Table 1.

### 2.3. Study Selection

Initially, the titles and abstracts of the studies identified by the search were screened independently by two authors (KMDC and LLL) using the Rayyan QCRI® tool (version 2016, Rayyan, Cambridge, CA, USA). Discrepancies were discussed with a third author (SHLV) until consensus was reached. A full-text review of all included papers was then completed to verify their eligibility.

### 2.4. Data Extraction

A spreadsheet was created to extract general information about each study (database, year of publication, authors, country of origin, type of design, objective). Information was also added on the neurodegenerative disease addressed, the target audience (patients, family members, caregivers), the route of feeding (oral or enteral), the educommunication actions or strategies (tools used, face-to-face or virtual) and the existing facilitating aspects or barriers (adherence, access to technology, level of education, dissemination, etc.). This information generated results that were tabulated and summarized in narrative form.

## 3. Results

The search strategy identified a total of 189 articles. After removing duplicates and excluding articles by reading the title and abstract, 29 articles were selected for reading the full text. Of these, three articles met the eligibility criteria and were included in this review. The detailed process followed the PRISM flow diagram (Figure 1).

Table 2 shows the characteristics of the three included studies focused on multiple sclerosis, dementia, and Parkinson’s disease, respectively. The first two studies were experimental prospective, and the last one was a mixed-methods experimental study.

One of the prospective studies aimed to assess the feasibility of a dietary intervention in multiple sclerosis patients, using a digital health approach in the research participants [26]. The other study aimed to investigate the effects of a dementia dietary educational program (DDEP) on caregivers’ nutritional knowledge, healthy eating behavior, and the nutritional status of people with dementia [27]. The mixed methods study aimed to increase knowledge about nutrition and improve the quality of life of people with PD and their caregiver’s using food and nutrition education mediated by technology [28].

In the study by Wingo et al. (2020) [26], the educational intervention consisted of materials sent weekly by email. Participants were instructed to record their daily food intake and exercise on a mobile app. The researchers used telecoaching to monitor progress and extract feedback from the participants. The authors mentioned that 90% of the participants completed the scheduled calls with the telecoach and that at least one meal was recorded on the mobile app in 82% of the days of the intervention. They concluded that telehealth was an effective method for conducting the study, as it enabled frequent contact with participants and avoided the challenges of scheduling face-to-face meetings [26].

In the study by Hsiao et al. (2020) [27], a nutrition education program for dementia was developed, based on the knowledge, attitude, and behavior model. Three phases were considered in nutrition education planning. Phase 1 was motivational, aimed at increasing awareness of the focal issues and stimulating change in associated behaviors. Phase 2 was an action improvement step, and phase 3 involved environmental factors and nutrition education methodological resources to support the teaching–learning process. This program consisted of six topics, each presented weekly. Each topic was covered in a separate session, each lasting 60 min. The topics were: altered eating behaviors in people with dementia, coping strategies for altered eating behaviors I and II, Mediterranean diet and dementia, nutritional assessment, and healthy food feast). In addition, health education leaflets and telecoaching were also strategies used in this program [27]. In this study, 69 pairs of people with dementia and their caregivers were recruited. Some participants were unable to continue with the study because of family reasons, personal factors, or death. At the end of the study, 57 pairs had completed the three phases, totaling 86.36% of the initial sample. The authors concluded that the dementia dietary educational program had positive effects, as both short- and long-term interventions effectively facilitated the acquisition of new knowledge and contributed to positive changes in eating behaviors related to food and nutrition during follow-up. In addition, it was identified that demographic factors played significant roles in the nutritional knowledge, healthy eating behavior, and nutritional status of people with dementia. For example, caregiver’s nutritional knowledge progressed among females and single caregivers significantly. In addition, the healthy eating behavior of the married caregivers improved significantly [27].

In the study by Brenes (2021) [28], the educational intervention followed a program similar to a Massive Open Online Course (MOOC). The program was applied individually and online for 8 weeks, using the Zoom video conferencing platform. This online format was used due to COVID-19 restrictions during the study period. The nutrition education plan consisted of six modules, with two video classes lasting between 5 and 15 min, as well as discussion forums. Self-determination theory was used to develop the program. This theory focuses on motivation, human personality, autonomy, competence, and relationships. In addition, digital leaflets related to the content of the video lessons and healthy recipes were made available to the participants [28]. In this study, out of a total of 28 participants, 54% completed the study and 50% had completed higher education. Participants answered pre- and post-test questions on quality of life and nutritional status, but there was no significant difference after the educational intervention. Motivation about nutritional knowledge increased after the intervention but was not significant. After the virtual program, 50% of the participants improved their quality of life. Despite the participants’ high interest and motivation, their effort to complete the program was high. This indicated the virtual program required a high effort or close attention to complete [28].

In the three studies included, the authors reported that the nutritional knowledge of participants improved after the educational intervention and had a positive effect on their eating behaviors and quality of life [26,28]. Information regarding facilitating aspects and barriers in the development and implementation of the educommunication actions proposed are found in Table 2.

## 4. Discussion

Of the 189 studies identified, only 3 used educommunication as a food and nutrition education strategy applied to caregivers and patients with neurodegenerative diseases. Although the other studies identified carried out nutritional education, only the studies by Wingo, et al. [26], Hsiao et al. [27], and Brenes [28] applied educommunication according to Soares’ conceptual perspective [29].

The studies of Wingo, et al. (2020) [26] and Brenes (2021) [28] carried out open and participatory communicative processes through tele-education, involving conferences, classes, and courses, using information and communication technologies [30], such as tele-coaching [26] and web conferencing [28]. On the other hand, the study of Hsiao et al. [27] used the modality of weekly face-to-face classes lasting 60 min. During the last session, each participant prepared a healthy meal and shared their experiences, promoting the empowerment and critical capacity of the participants involved, contributing significantly to learning. In addition, the authors also used educational leaflets and telecoaching.

Educommunication integrates two strategic areas for learning: education and communication. This area considers that the interaction between education and communication favors the development of innovative and enriching pedagogical actions, therefore it proposes the exchange of knowledge and the empowerment of the protagonist in the learning process [19,31,32,33,34]. The World Health Organization recommends nutritional education programs as a form of health promotion, using strategies to improve learning, skills, and self-care in food choices [35,36].

In the systematic review conducted by Atoloye et al. [37], despite the absence of studies involving patients with neurodegenerative diseases and caregivers, the findings align with those of Wingo, et al., Hsiao et al., and Brenes [26,27,28] regarding the efficacy of immediate nutritional intervention in fostering acquired knowledge and inducing positive changes in diet and nutrition. This highlights the importance of nutritional education as a contributing factor to raising awareness and promoting health improvements for patients with neurodegenerative diseases.

However, food and nutrition education strategies are often not properly adapted to meet the needs of patients and their caregivers and may not generate significant results in changing nutrition-related behavior or quality of life [27]. The participation and practical involvement of the users who receive the training or who consume the audiovisual content is essential to generate active and functional learning. For this reason, the educational program must be designed considering the healthcare scenario and the recipients, among other essential elements of communication [38,39,40].

Educommunication can play a fundamental role in overcoming this problem, as it helps to simplify complex concepts and ideas, making it more accessible and understandable through an approach that involves effective communication. Simplifying complex concepts does not always mean sacrificing scientific rigor. It means adjusting the scientific message to fit mass media standards, encouraging dialogue and learning in health [17]. In this process, complex concepts are explained in detail and simplified in order to adapt them to the pedagogical process. The strategic use of media to educate, raise awareness, and empower individuals integrates two fundamental areas: education and communication from an interdisciplinary perspective. The innovative idea of educommunication establishes a new paradigm for education, exerting a direct influence on the development of teaching methods [14,29]. Although there is a limited number of studies on this subject, integrating educommunication into food and nutrition education can significantly impact the teaching of caregivers and patients with neurodegenerative diseases, thus enhancing their survival and quality of life [3]. This education can extend beyond current social media communication trends by focusing on specialized content tailored for each platform, reflecting the individualized nature of targeted audiences today.

Food and nutrition education for neurodegenerative diseases is challenging due to several factors, including cognitive impairment, motor function issues, communication barriers, the progressive nature of the diseases, individualized needs, and resource limitations. Consequently, caregivers play a crucial role in managing the patient’s diet and nutritional status. Additionally, educommunication strategies tailored to specific neurodegenerative disease groups or stages of the disease are essential for effective learning and improved outcomes.

Therefore, educommunication is a concept that goes beyond the boundaries of formal and non-formal education. It advocates for a dialogical approach, utilizing communication as a tool for reflection rather than just for transmitting content. In this view, education is not confined to a pedagogical and psychological process of acquiring knowledge; it is also a way of expanding perspectives, affecting methods of socialization and communication, valuing reality, and shaping behavioral patterns and attitudes [19]. Thus, educommunication strategies on food and nutrition education can also play a preventive role, given the strong connection between diet and neurodegenerative diseases.

This is the first review in the literature to map educommunication strategies for food and nutrition education targeting caregivers and patients with neurological diseases. Its originality contributes significantly to the existing body of knowledge in this field. Additionally, this review identifies both facilitating factors and barriers in the development and implementation of educommunication within the context of neurodegenerative diseases. This information highlights the importance of educommunication as a strategy to improve nutritional status, prevent malnutrition, and slow the progression of these diseases. It also emphasizes the need for well-planned strategies in food and nutrition education to achieve effective outcomes.

The number of studies included in this review was a limiting factor. Restricting the search to neurodegenerative diseases and working with a conceptual perspective on educommunication may have reduced the number of included studies. On the other hand, studies involving nutritional education interventions need to have their methodologies more thoroughly described both to enhance reader understanding of the use of educommunication and for purposes of reproducibility.

## 5. Conclusions

According to the studies included in this review, the main educommunication strategies used for food and nutrition education were educational modules delivered by email, telecoaching, telephone counseling, web conferencing, mobile application to record food intake, online focus groups, and a blended learning strategy, including demonstration, training, and self-monitoring. For the development and implementation of educommunication, the main facilitating aspects were the use of telehealth, telecoaching, hands-on activities, topics tailored to participants’ interests and needs, a high education level of the participants as well as easy access to electronic devices. The main barriers identified were daily journaling to track food intake, specific demographic factors influencing engagement and the learning process, lack of access to electronic devices, and missing classes due to personal situations. Some authors comment that short duration, small sample size, and the absence of a control group may have influenced the learning effects. Despite these barriers, the educommunication strategies implemented facilitated the acquisition of knowledge in food and nutrition, resulting in health benefits for the participants. Studies using educommunication strategies for food and nutrition education in neurodegenerative diseases are still scarce. It is essential to incorporate educommunication strategies into food and nutrition programs. We believe this review can provide valuable insights for future research needed in this field.

## Figures and Tables

**Figure 1 ijerph-21-01113-f001:**
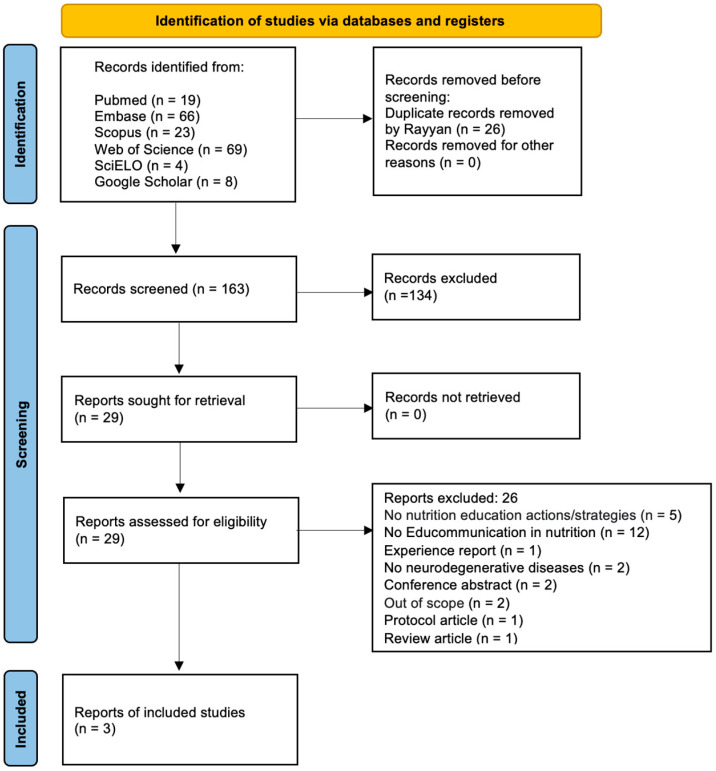
Flow diagram showing the scoping review searching and screening processes.

**Table 1 ijerph-21-01113-t001:** Search strategies in databases.

Database	Equations
PubMed	(neurodegenerative disease [MeSH Terms]) AND (nutritional sciences [MeSH Terms] OR nutritional status [MeSH Terms]) AND (“education” [MeSH Terms] OR “literacy” [MeSH Terms] OR “educommunication” [All Fields] OR Teach-Back Communication)
Embase	‘degenerative disease’/exp AND ‘nutrition education’/exp
Scopus	(“Neurodegenerative Diseases” OR “Degenerative Diseases Central Nervous System” OR “Degenerative Diseases Nervous System”) AND (“Nutrition Sciences” OR “Nutrition”) AND (“Education” OR “Literacy” OR “Educommunication” OR “Teach-Back Communication”)
Web of Science	(“Neurodegenerative Diseases” OR “Degenerative Diseases Central Nervous System” OR “Degenerative Diseases Nervous System”) AND (“Nutrition Sciences” OR “Nutrition”) AND (“Education” OR “Literacy” OR “Educommunication” OR “Teach-Back Communication”)

**Table 2 ijerph-21-01113-t002:** Characteristics of the included studies in this scoping review.

Authors (Year) [Ref] Country	Study Type	Study Aim	Study Population	Feeding Route	Educommunication Actions and Strategies	Type of Approach	Facilitating Aspects	Barriers	Conclusion
Wingo et al. (2020) [23] United States	Experimental prospective	To test the feasibility of delivering a low glycemic index (GL) dietary intervention implemented via telehealth in a sample of adults with relapsing remitting MS	Patients with multiple sclerosis (MS) (*n* = 22)	Oral	The intervention consisted of three main components: weekly modules delivered by email, weekly contacts with a telecoach, and daily use of a mobile application to record food intake and physical activity	Hybrid (in person and online)	The individualized instruction via telehealth and the weekly telecoach contact contributed to increasing the level of nutrition knowledge, improving dietary adherence, and keeping the high retention rate in the study	Journaling daily in the long term was challenging, even with the use of a mobile app. The absence of a control group may have hidden other results regarding the intervention, since the participants included were people behaviorally ready for a diet change	A low GL dietary intervention is feasible for adults with relapsing remitting MS and may lead to improvements in MS outcomes and cardiometabolic risk
Hsiao et al. (2020) [24] Taiwan	Experimental prospective	To investigate the effects of a dementia dietary educational program (DDEP) on family caregivers’ nutritional knowledge, healthy eating behavior, and nutritional status of patients with dementia	Patients with dementia and caregivers (*n* = 57 pairs), divided in control group and experimental group	Oral	DDEP adopted a blended learning strategy and combined skills of demonstration, ability training in assessment, and self-monitoring. The DDEP consisted of six topics about the Mediterranean diet, management of eating problems, and nutrition care. Each topic was presented weekly in 60 min. sessions. Group A received routine care with health education leaflets regarding nutritional knowledge for dementia, and telephone counseling every 2 weeks. Group B received the DDEP, telephone counseling every 2 weeks, and care tracking	Hybrid (in person and online)	Various teaching and idea-sharing strategies were used during the intervention period. Hands-on activities on meal preparation allowed participants to share their experiences and be directly involved in their learning. The DDEP had more potential long-term effects on healthy eating and behavior change compared to routine care	Small sample size, demographic factors (age, gender, marital status, employment, level of education), and missing sections may have influenced the learning effects	DDEP had greater potential for long-term effects on nutritional knowledge and healthy eating. When designing food and nutrition education intervention, the demographic characteristics of participants and cognitive function of the patients must be considered to provide a more tailored approach
Brenes (2021) [25] United States	Experimental mixed methods	To increase nutritional knowledge and improve quality of life through a virtual nutrition education program to people with PD and caregivers	Patients with Parkinson’s disease (PD) and caregivers (*n* = 28)	NR	A virtual program on nutritional education was divided into six modules. Used focus groups to determine topics of interest and pre- and post-self-report data to assess program effectiveness and quality of life. Used web conferencing for online meetings	Online	The program covered topics the participants were interested in. The virtual program was easy to understand and navigate. Most participants had high education and possibly higher socioeconomic levels which facilitated the comprehension, and the access to special foods and electronic devices to complete the virtual program	The study’s short duration, small sample size, absence of a control group, and the fact it was conducted during the COVID-19 pandemic may have affected some results. The effort to complete the program was high and not all participants completed all surveys. Participants of this study are not representative of the whole PD population since people without access to electronic devices and internet services could not enroll in the program	This virtual program was tailored specifically to the knowledge needs of the participants. It helped improve the quality of life by providing support through nutrition knowledge

DDEP, dementia dietary educational program. GL, glycemic index. MS, multiple sclerosis. NR, not reported. PD, Parkinson’s disease.

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
