# Peer review of "Educommunication in Nutrition and Neurodegenerative Diseases: A Scoping Review"

_ijerph, 2024, doi:10.3390/ijerph21081113_

Round 1

Reviewer 1 Report

Comments and Suggestions for Authors

Dear authors, 

I suggest some suggestions.

I think the tittle of the article should be other because content of the the review is more closed to the use of blended learning strategies, behavourism strategies or information and communication technology (ICT) than to the educomununicational strategies. 

Here you have some questions about some afirmations:

194. What is the Soare´s conceptual perspective they applied? There isn-t any reference to it in the introduction

134-135. In the educommunication the person is the protagonist. In this case, the experience is closed to the behaviorism: "Participants were instructed"

253-259: the strategies are educommunicational strategies? Are the people protagonist of their process as the educomunicatios¡n says?

In the Discussion, most of the references the authors used are not explained at introduction.

Author Response

Dear Reviewer 1,

Thank you for reviewing our manuscript. Your thoughtful comments and constructive feedback have been invaluable in refining and improving the quality of our work.

Please find attached a detailed point-by-point response to your comments.

Best Regards,

Lucia Leite-Lais

Reviewer 2 Report

Comments and Suggestions for Authors

The authors should also provide more scientific information on the link between nutrition and neurodegenerative diseases in the introduction section to strengthen the overall impact of the paper. Currently, there is only one short sentence addressing this link. After the paragraph describing neurodegenerative diseases, or within the same paragraph, authors should add more information on how nutrition affects the progression of these diseases and the quality of life of affected individuals.

The paper cites several other systemic reviews, but the authors need to clarify how this review contributes to the existing body of knowledge beyond just summarizing three studies. 

In the results section, authors should aim to describe each study in cohesive section rather than jumping back and forth as it disrupts the flow of information..

Given the complexity of neurodegenerative diseases, which can manifest differently depending on the specific context, the authors should further elaborate on the knowledge gaps and suggest strategies, if any, that may be used tailored to specific diseases. For instance, dementia patients with memory loss may benefit from different educommunication strategies compared to neurodegenerative patients experiencing issues primarily with their motor skills. Educommunication programs need to be customized based on the specific neurodegenerative disease groups to ensure patients can retain as much information as possible. I suggest that the authors mention information of this sort in the discussion/conclusion section.

I suggest that authors also briefly discuss in the conclusion section how such programs can be used in older asymptomatic/healthy adults as a preventative measure to delay the onset of neurodegenerative diseases, given the strong link of diet with neurodegenerative disease.

Comments on the Quality of English Language

The paper is well written but I recommend that the authors proof-read the paper for minor grammatical errors such as lack of conjunctions (eg. and when listing multiple things), wrong/inconsistent use of hyphenation).

There are also some undefined abbreviations or abbreviations not defined at first use (eg. NR, GL etc...).

Author Response

Dear Reviewer 2,

Thank you for reviewing our manuscript. Your thoughtful comments and constructive feedback have been invaluable in refining and improving the quality of our work.

Please find attached a detailed point-by-point response to your comments.

Best Regards,

Lucia Leite-Lais

Reviewer 3 Report

Comments and Suggestions for Authors

Dear Authors

I reviewed your manuscript, which touches on important aspects of education. 

The manuscript is well written, according to actual recommendations, but it is challenging to follow and read. It needs rephrasing to gain better readability.

In general, this article needs more work to improve it. The whole body of the manuscript needs rephrasing. 

My detailed suggestions are pointed out below:

- in the introduction section, please explain in detail what educommunication is and why it is different from other kinds of communication in medical education.

- Inclusion and exclusion criteria.

Based on PRISMA guidelines, please add information about comparison and outcomes. Why do you don't use it? Please add information and explain.

- Results section

Starts from line 126,  in the first few paragraphs you described analyzed papers, but it isn't easy to follow which one is mentioned below. Of course, the reader may check references each time, but it takes work to follow. 

Comments on the Quality of English Language

Some grammar and style correction are needed. 

Author Response

Dear Reviewer 3,

Thank you for reviewing our manuscript. Your thoughtful comments and constructive feedback have been invaluable in refining and improving the quality of our work.

Please find attached a detailed point-by-point response to your comments.

Best Regards,

Lucia Leite-Lais

Reviewer 4 Report

Comments and Suggestions for Authors

Comments

1-     Lines between 77-82: need more details about sites, methods, and all mentioned links, for me I spent time considered waste time to revise all of them.

2-     Table-2: Tables in scientific papers should be self- explanatory; so all abbreviations to be clarified- as GL, MS, and PD- at least under the mentioned table.

3-     Some of references to be clearly cited like:[Brenes, P. Virtual Nutrition Education for People Affected by Parkinson’s Disease, College of Health and Human 347 Sciences: Manhattan, 2021]

Author Response

Dear Reviewer 4,

Thank you for reviewing our manuscript. Your thoughtful comments and constructive feedback have been invaluable in refining and improving the quality of our work.

Please find attached a detailed point-by-point response to your comments.

Best Regards,

Lucia Leite-Lais

Round 2

Reviewer 1 Report

Comments and Suggestions for Authors

Dear authors, 

Comment and Response 1, 2, 3 & 4: I consider that, as I commented in the first review, the title of the article should be different because the content of the review is more closed to the use of blended learning strategies, behaviourist strategies or information and communication technologies (ICT) than to the use of educommunicative strategies. 

The educommunicative approach refers not only to the use of ICT in education, but to a deeper integration between educational and communicative processes, promoting active participation, dialogue and collaborative production of content. Furthermore, educommunication emphasises the development of critical thinking and the ability of individuals to analyse and produce media messages. A behaviourist approach, which focuses on conditioning and response, may not fully align with these goals if it is not complemented by practices that encourage critical reflection and participation.  Educommunication also involves democratic media management, which means that participants have an active role in the creation and dissemination of content, beyond being passive recipients. For educommunication to be said to be implemented, the programme should incorporate these key principles, not just the use of technology and behavioural methods.

In the introduction of the paper, you explain some of the characteristics of the educommunication according to Soare:

Line 69-70: "This methodology seeks to facilitate social dialogue". 

Line 78-80: "to expand the awareness and critical participation of individuals in communicative ecosystems". 

Line 89-90: "promoting discussions and exchanges between participants. This is differ90 ent from traditional medical education".  

Line 93-94: "by enhancing their ability to express themselves and participate actively in their  learning process."

In the review, however, the studies are close to behaviourism and exogenous models:

Line 173: "In the study by Wingo et al. (2020) [23][26], the educational intervention consisted of materials sent weekly by email. Participants were instructed to record their daily food intake and exercise on a mobile app". Any references to the characteristics of the educommunication. 

Line 182: In the study by Hsiao et al. (2020) [24][27], a nutrition education program for dementia was developed, based on the knowledge, attitude, and behavior model. Any references to the characteristics of the educommunication.

Line 225: "discussion forums". This can be an example of educommunicative practice by including discussion forums.

The three studies included in the review use information and communication technologies but their strategies cannot be considered educommunicative. 

Comment and Response 5: It is ok

The focus of the article and its title should be changed to an approach based on the use of information and communication technologies in educational programmes and blended learning approaches.

Author Response

Thank you again for your comments. Please see our answer attached. 

Reviewer 3 Report

Comments and Suggestions for Authors

Dear Authors thank you for your effort. I see you revise it so much and I have only one concern. In section 2.1. Inclusion and exclusion criteria add information on what kind of guideline do you base. If you are workinf on different one, please be more specify. 

Author Response

Thank you for your additional comment. Please see our answer attached.
